# The Change in Glucagon Following Meal Ingestion Is Associated with Glycemic Control, but Not with Incretin, in People with Diabetes

**DOI:** 10.3390/jcm10112487

**Published:** 2021-06-04

**Authors:** Soyeon Yoo, Dongkyu Kim, Gwanpyo Koh

**Affiliations:** 1Department of Internal Medicine, College of Medicine, Jeju National University, Jeju 63241, Korea; happyweed@jejunu.ac.kr; 2Department of Internal Medicine, Jeju National University Hospital, Jeju 63241, Korea; 3Department of Medicine, Graduate School of Jeju National University, Jeju 63243, Korea; kdkjeju@hanmail.net

**Keywords:** diabetes mellitus, glucagon, glucagon-like peptide 1, glucose-dependent insulinotropic polypeptide

## Abstract

Background: We aimed to investigate the changes in glucagon levels in people with diabetes after the ingestion of a mixed meal and the correlations of variation in glucagon levels with incretin and clinico-biochemical characteristics. Methods: Glucose, C-peptide, glucagon, intact glucagon-like peptide 1 (iGLP-1), and intact glucose-dependent insulinotropic polypeptide (iGIP) were measured in blood samples collected from 317 people with diabetes before and 30 min after the ingestion of a standard mixed meal. The delta (Δ) is the 30-min value minus the basal value. Results: At 30 min after meal ingestion, the glucagon level showed no difference relative to the basal value, whereas glucose, C-peptide, iGLP-1, and iGIP levels showed a significant increase. In univariate analysis, Δglucagon showed not only a strong correlation with HbA1c but also a significant correlation with fasting glucose, Δglucose, and estimated glomerular filtration rate. However, Δglucagon showed no significant correlations with ΔiGLP-1 and ΔiGIP. In the hierarchical multiple regression analysis, HbA1c was the only variable that continued to show the most significant correlation with Δglucagon. Conclusions: People with diabetes showed no suppression of glucagon secretion after meal ingestion. Patients with poorer glycemic control may show greater increase in postprandial glucagon level, and this does not appear to be mediated by incretin.

## 1. Introduction

Glucagon, which is secreted by the α-cells of the islet of Langerhans, plays an important role in maintaining glucose homeostasis along with insulin, which is secreted from the β-cells of the pancreas. In hyperglycemia, insulin secretion is increased, and glucagon secretion is suppressed, whereas, in hypoglycemia, insulin secretion is inhibited, and glucagon secretion is increased, causing an increase in hepatic glucose production. Blood glucose levels are maintained at normal levels through the completely contrasting actions of insulin and glucagon [1]. Glucagon was first discovered in 1921 from an extract of animal pancreas and is known to increase the blood glucose level. However, it has not gained much attention, unlike insulin, which has a therapeutic effect for diabetes [2]. In 1973, Unger and Orci presented the bihormonal-abnormality hypothesis, which states that excess glucagon could play an important role equivalent to that of insulin deficiency in the pathogenesis of diabetes [3,4]. Subsequently, many studies on glucagon were actively published.

After ingesting a meal, healthy adults show increased insulin secretion and suppressed glucagon secretion, whereas people with diabetes do not show suppression of glucagon secretion [5,6]. Knop et al. [7] reported that healthy individuals showed suppression of glucagon in both the oral glucose tolerance test (OGTT) and isoglycemic intravenous glucose infusion (IIGI), whereas people with type 2 diabetes (T2D) showed suppression of glucagon only in IIGI but not in OGTT. Muscelli et al. [8] and Oh et al. [9] reported inappropriate suppression of glucagon secretion in OGTT among T2D patients, unlike in healthy individuals. The mixed meal tolerance test (MMTT) is a more physiological test than OGTT in assessing the dynamic response to insulin or glucagon secretion. After ingesting a mixed meal, blood glucagon levels become significantly higher in people with diabetes than in healthy individuals, and the peak level is reached within 30 min after ingestion [10,11]. The phenomenon of unsuppressed glucagon secretion only after oral ingestion appears not only in T2D but also in all types of diabetes including type 1 diabetes [12], diabetes secondary to chronic pancreatitis [13] or pancreatectomy [14], and maturity-onset diabetes of the young [15], among various other types [16]. The mechanisms that could explain such an inappropriate postprandial glucagon response among people with diabetes include resistance of α-cells to insulin or glucose [16], decreased insulin secretion [17], increased ratio of α-/β-cells in the islets of Langerhans [18,19], and gut-origin glucagon [20], but the exact mechanism is still not clearly understood.

Incretin is a group of hormones secreted by the enteroendocrine cells after ingestion of food, and it stimulates β-cells to increase insulin secretion, which, in turn, prevents an increase in the postprandial blood glucose level. Glucagon-like peptide 1 (GLP-1) and glucose-dependent insulinotropic polypeptide (GIP) are typical incretins that are secreted from the L-cells of the ileum and K-cells of the duodenum and jejunum, respectively [21]. GLP-1 and GIP have different regulatory effects on insulin and glucagon secretion. GLP-1 stimulates insulin secretion but suppresses glucagon secretion [22], whereas GIP stimulates both insulin and glucagon secretion [23,24]. Therefore, gut-derived hormones such as incretins may mediate the paradoxical glucagon response that appears when food is ingested orally [7,25].

The mechanism underlying the inappropriate suppression of glucagon secretion that appears after food ingestion in diabetes patients has not yet been identified. To the best of our knowledge, there are almost no studies that have analyzed changes in glucagon levels after the ingestion of a mixed meal in a large patient population. Accordingly, we aimed to investigate the changes in glucagon levels in approximately 300 diabetes patients after the ingestion of a standard mixed meal and evaluate the correlations of these changes with various clinico-biochemical variables, including incretins, to identify the factors influencing the inappropriate suppression of glucagon secretion.

## 2. Materials and Methods

### 2.1. Subjects

The study population comprised 317 people with diabetes who had visited the diabetes clinic at the Department of Endocrinology and Metabolism, Jeju National University Hospital, Patients with stage 3 or 4 chronic kidney disease (estimated glomerular filtration rate (eGFR) <30 mL/min), liver cirrhosis with ascites, or an infectious or inflammatory disease were excluded. Patients taking dipeptidyl peptidase-4 (DPP-4) inhibitors or GLP-1 receptor agonists, which can affect blood glucagon and incretin levels, were also excluded. This study was conducted in accordance with the Declaration of Helsinki and was approved by the Institutional Review Board (IRB) of Jeju National University Hospital (IRB File No. JEJUNUH 2020-07-018). Written informed consent was obtained from all subjects before participation in the study. 

### 2.2. Study Procedure

Blood samples were collected after 8 h of overnight fasting and were stored in heparinized tubes and chilled ethylenediaminetetraacetic acid (EDTA) tubes containing aprotinin (250 KIU/mL blood; Sigma-Aldrich, St Louis, MO, USA) and DPP-4 inhibitor (10 μL/mL blood; Merck Millipore, Darmstadt, Germany). Glucose, hemoglobin A1c (HbA1c), C-peptide, creatinine, and alanine aminotransferase (ALT) levels were measured from the blood samples stored in heparinized tubes, while fasting intact GLP-1 (iGLP-1), intact GIP (iGIP), and glucagon levels were measured from the blood samples stored in chilled EDTA tubes containing aprotinin and DPP-4 inhibitor. Subsequently, the subjects were provided with a standard mixed meal comprising rice, soup, three side dishes, and kimchi. The mixed meal provided to each subject always had a consistent number of calories (480 kcal) and composition ratio of macronutrients (carbohydrate:protein:fat = 2.8:1:1). The subjects ate the mixed meal within 20 min, and blood samples were collected again at 30 min after the start of the meal and were stored in heparinized tubes and chilled EDTA tubes containing aprotinin and DPP-4 inhibitor. The blood samples stored in heparinized tubes were used to measure 30-min postprandial glucose and C-peptide levels, and those stored in chilled EDTA tubes were used to measure 30-min postprandial iGLP-1, iGIP, and glucagon levels. The EDTA tubes containing blood samples were chilled and stored at −20 °C until iGLP-1, iGIP, and glucagon levels were measured. The difference between the 30-min postprandial value and the basal value (Δ, 30-min minus basal) for glucose, C-peptide, iGLP-1, iGIP, and glucagon was expressed as Δglucose, ΔC-peptide, ΔiGLP-1, ΔiGIP, and Δglucagon, respectively. 

Information regarding age, duration of diabetes (DM duration), other medical history, and history of diabetes medication was obtained by directly asking the subject or by referring to electronic medical records from Jeju National University Hospital. Height and weight were measured using an electronic device, while blood pressure was measured twice, each after at least 10 min of rest, and the mean value from two measurements was recorded. Waist circumference (expressed in centimeters, rounded to one decimal place) was measured mid-breath with the legs placed shoulder-width apart, measuring from the midpoint between the last rib and the iliac crest parallel to the ground.

### 2.3. Measurements of Biochemical Markers

Plasma glucose levels were measured by the glucose oxidase method using the TBA-200FR chemical analyzer (Toshiba, Tokyo, Japan). HbA1c levels were measured by ion-exchange high-performance liquid chromatography using HLC-723G8 (Tosoh, South San Francisco, CA, USA). C-peptide levels were measured using Modular Analytics E170 electrochemiluminescence immunoassays (Hitachi, Tokyo, Japan). Creatinine and ALT levels were measured using the TBA-200FR chemical analyzer, while eGFR was calculated using the Modification of Diet in Renal Disease equation [26]. For the quantification of iGLP-1 (GLP-1 (7–36) amide, GLP-1 (7–37)) and iGIP (GIP (1–42)), a measurement kit based on the principle of sandwich enzyme immunoassay (Code Nos. 27784 and 27201, respectively; Immuno-Biological Laboratories Co. Ltd., Gunma, Japan) was purchased and used. For the quantification of glucagon, a glucagon chemiluminescent kit based on sandwich enzyme-linked immunosorbent assay (Cat. No. EZGLU-30K, Merck KGaA, Darmstadt, Germany) was purchased and used. 

### 2.4. Statistical Analysis

All values are expressed as mean ± standard deviation or percentage. The differences between 30-min postprandial and fasting glucose, C-peptide, iGLP-1, iGIP, and glucagon levels were tested using the Wilcoxon signed-rank test. Correlations between clinico-biochemical variables and Δglucagon were analyzed using the Pearson’s correlation analysis, while differences in Δglucagon according to categorical variables were analyzed using the Mann–Whitney *U* test. The degree of stepwise contributions of various clinico-biochemical variables on Δglucagon was assessed using hierarchical regression analysis. For the analyses, Δglucagon was set as the dependent variable, while the independent variables varied for each model. Model 1 included age, sex, fasting glucose, C-peptide, and factors that showed significant correlations in Pearson’s correlation analyses. In Model 2, Δglucose, ΔC-peptide, ΔiGLP-1, and ΔiGIP were added. The final model (Model 3) included diabetes medication that showed significant differences in Δglucagon. Before performing Pearson’s correlation analysis and hierarchical regression analysis, all variables showing non-normal distribution were logarithmically transformed. All statistical analyses were performed using SPSS 14.0 (SPSS Inc., Chicago, IL, USA), and *p* <0.05 was considered to indicate statistical significance. 

## 3. Results

### 3.1. Clinico-Biochemical Characteristics of Subjects and Ingestion of Mixed Meal Induce Increase Incretin Level, but Not Glucagon Levels

Based on the inclusion and exclusion criteria applied, the study population comprised a total of 317 patients, and the clinico-biochemical characteristics are shown in Table 1. Most of the patients were middle-aged or elderly, and there were slightly more males than females. DM duration was relatively long, and glycemic control tended to be poor. Approximately two-thirds of the patients were taking metformin, half of them were taking sulfonylurea, and one-third were taking insulin (Table 1). 

Table 2 shows the glucose, C-peptide, iGLP-1, iGIP, and glucagon levels of the patients before and after the ingestion of the mixed meal. The results show a significant increase in 30-min postprandial glucose, C-peptide, iGLP-1, and iGIP levels relative to the basal levels. Glucagon secretion was not suppressed by the ingestion of the mixed meal, and there was no difference between 30-min postprandial and basal levels (Table 2 and Figure 1A).

### 3.2. In Univariate Analysis, ΔGlucagon Levels Showed Correlation with HbA1c, Fasting Glucose, Δglucose, and GFR, but Not Incretin Levels

Pearson’s correlation analysis and Mann–Whitney *U* test were performed as univariate analyses to investigate the correlation between the degree of unsuppressed glucagon secretion after meal ingestion and clinico-biochemical variables. In Pearson’s correlation analysis for continuous variables, Δglucagon showed significant positive correlations with HbA1c and fasting glucose and significant negative correlations with Δglucose and eGFR (Table 3). In particular, the HbA1c level showed the highest correlation with Δglucagon (*r* = 0.389) (Figure 1B), which indicates that glucagon levels actually increase after a meal instead of being suppressed when glycemic control is poor. Changes in glucagon levels after the ingestion of the mixed meal according to sex and medication used were compared between groups using the Mann–Whitney *U* test. The margin of increase in glucagon levels was significantly larger in females than in males (−6.4 ± 45.7 vs. 11.5 ± 49.5, *p* < 0.001). Moreover, the margin of increase in glucagon was significantly smaller in patients who use sulfonylurea (−5.9 ± 41.8 vs. 6.6 ± 53.2, *p* = 0.017) and metformin (−4.3 ± 41.1 vs. 9.7 ± 59.6, *p* = 0.014) and significantly larger in patients who use insulin than in those who do not (11.1 ± 49.5 vs. −5.2 ± 49.2, *p* < 0.001). 

### 3.3. In Hierarchical Multiple Regression Analysis, HbA1c Was the Variable Predicting ΔGlucagon Levels

Hierarchical multiple regression analysis was performed to identify the independent predictors of the margin of increase in the postprandial glucagon level with exclusion of the confounding effects of other variables. In Model 1, which included age, sex, and factors that showed significance in univariate analysis (HbA1c, eGFR, and glucose), sex (female) and HbA1c showed a significant positive correlation with Δglucagon. In Model 2, the margin of increase in 30-min postprandial glucose, C-peptide, iGLP-1, and iGIP was added as a variable to Model 2 to analyze the correlations with Δglucagon. The explanatory power (adjusted R^2^) of Model 2 increased by 6.8%, as compared to that of Model 1. In Model 2, the HbA1c level continued to show a significant correlation with Δglucagon, whereas sex did not show a significant correlation. The margin of increase in glucose levels showed a significant negative correlation with Δglucagon, but iGLP-1 and iGIP levels did not show significant correlations. In Model 3, which include diabetes medication as additional variables, the explanatory power (adjusted R^2^) increased by 0.7%, as compared to that of Model 2, while HbA1c, Δglucose, and sex showed significant correlations with Δglucagon. In all three models, the corrected regression coefficient (*β*) of HbA1c was highest, based on which it was determined that glycemic control is the best predictor of the increase in the postprandial glucagon level (Table 4). 

## 4. Discussion

Analyses of 317 diabetes patients in our study showed that the margin of increase in glucagon levels after the ingestion of a standard meal was significantly positively correlated with HbA1c, significantly negatively correlated with the increase in postprandial blood glucose levels, and had no correlation with iGLP-1 and iGIP. Therefore, it could be concluded that the postprandial glucagon level increased more with poorer glycemic control, while incretin, which is known to control glucagon secretion, was not correlated with changes in the postprandial glucagon level. In other words, blood glucose level increased 30 min after the ingestion of food in diabetes patients, but glucagon secretion was not suppressed; contrary to expectations, it may actually increase in patients with poor glycemic control. The results also show that such changes in glucagon levels were not mediated by GLP-1 or GIP.

However, the reason for the increase in postprandial glucagon levels being significantly higher in patients with poor glycemic control in our study is not clearly known. It may be due to the postprandial glucagon suppression not appearing when insulin secretion decreases. As the duration of T2D increases, the blood glucose level is gradually elevated, as there is a gradual reduction in the ability of β-cells to secrete insulin [27]. Moreover, insulin deficiency has been presented as the most important mechanism involved in elevated blood glucagon level [17,28,29]. Therefore, it could be explained that HbA1c and the increase in postprandial glucagon showed positive correlation in our study because elevated HbA1c indicates reduced ability to secrete insulin. However, DM duration, which is known to show an inversely proportional relationship with insulin secretion [30], showing no correlation with the margin of increase in the glucagon level in our study, does not support the explanation above. Moreover, poor glycemic control itself could be the cause of inappropriate suppression of glucagon. This finding is in line with those reported by Aydin et al. [31] and Raskin et al. [32]. These studies confirmed that insulin administration in diabetes patients resulted in improved control of blood glucose level and decreased blood glucagon level. However, it is not certain whether the decrease in blood glucagon level is due to improved blood glucose level or increased blood insulin level. On the contrary, inappropriate suppression of glucagon secretion could be the cause of poor glycemic control. Increased glucagon increases hepatic glucose production, which can lead to hyperglycemia. In a study conducted in normal subjects, hepatic glucose output was completely suppressed within 30 min of mixed meal ingestion. Subsequently, it gradually increased between 60 and 255 min, and then rapidly recovered to the basal level [33]. The change in hepatic glucose output induced by mixed meal was consistent with the change in plasma glucagon/insulin ratio. The suppression of hepatic glucose production by mixed meal intake, as well as the decrease in postprandial glucagon/insulin, was lower in T2D patients than in healthy individuals [34]. Since hepatic glucose output was not directly measured in this study, a clear causal relationship could not be established. However, correlation analysis between the glucagon/insulin ratio and HbA1C in patients who did not take insulin among the subjects of this study (Appendix A), similar to Δglucagon, revealed that Δglucagon/insulin ratio was significantly positively correlated with HbA1c and had no correlation with iGLP-1 and iGIP. As another mechanism by which an increase in glucagon may cause an increase in blood glucose, Patarrão et al. [35] previously reported that glucagon decreases hepatic GSH and increases insulin resistance. Nevertheless, in this study’s subjects who were not exposed to insulin, HOMA IR (%), an index of insulin resistance, did not affect the association between Δglucagon and HbA1C (Appendix A). In conclusion, the positive correlation between HbA1c and increase in postprandial glucagon observed in our study may be attributed to the following reasons: postprandial glucagon not being suppressed due to reduced insulin secretion, poor glycemic control itself causing inappropriate suppression of glucagon, and inappropriate postprandial suppression of glucagon itself causing poor glycemic control, by not effectively suppressing hepatic glucose output. However, the exact mechanism could not be identified based on the findings in our study owing to the use of the cross-sectional design.

The results of multivariate analysis show a negative correlation between Δglucose and Δglucagon, which could be explained by the well-known opposing actions of glucose and glucagon. It is believed that the increase in blood glucose level caused by the ingestion of food suppressed glucagon secretion. The results of our study do not show a significant correlation between ΔC-peptide and Δglucagon. C-peptide is an indirect indicator of insulin [1], which is known to suppress glucagon secretion. Therefore, the aforementioned results may appear dubious, but this may have been caused by the collection of blood samples at only 30 min after the ingestion of a mixed meal. People with diabetes have reduced β-cell functions, as compared to healthy individuals; therefore, the peak level of C-peptide appears 90–120 min after the ingestion of food [36]. Therefore, if blood samples were collected at 90–120 min after the meal, instead of 30 min, then there may have been a significant correlation between ΔC-peptide and Δglucagon. 

Intravenous infusion of GLP-1 suppresses glucagon secretion [22], whereas intravenous infusion of GIP stimulates glucagon secretion [23,24]. Accordingly, changes in glucagon levels in our study were expected to show a negative correlation with changes in iGLP-1 and a positive correlation with changes in iGIP, but such results were not found. Similarly, other studies have also reported no correlation between changes in postprandial glucagon and changes in incretin levels. Knop et al. [7] reported that OGTT performed on healthy individuals and diabetes patients showed difference in glucagon suppression, whereas there was no difference between the groups regarding response to GLP-1 and GIP. Yabe et al. [11] performed OGTT and MMTT on healthy individuals, patients with impaired glucose tolerance, and diabetes patients and concluded that glucagon is not associated with response to incretin based on the findings that glucagon was increased to a greater extent in diabetes patients, whereas there was no difference in GLP-1 and GIP levels among the three groups. In other words, considering the findings of our study as well as the results of other studies, it appears that changes in glucagon levels appearing after a meal in people with diabetes are not caused by endogenous GLP-1 or GIP.

The results of univariate analysis performed in our study show that the margin of increase in glucagon levels after the ingestion of the mixed meal was larger among females than among males, while the results of multivariate analysis also confirmed that female sex is a significant predictor of the increase in glucagon levels. Horie et al. [37] also reported that females showed a lesser extent of suppression of glucagon secretion than males in OGTT. When various types of stimulation, including hypoglycemia, were applied in a study using mice, female mice showed greater increase in blood glucagon levels than male mice [38]. Although the exact reason greater glucagon response is found among females, in both humans and animals, is unknown, it has been explained because of the following reason: females have larger pancreatic α-cell mass than males [39].

Our findings are also in line with those of another study that was recently published. Similar to our study, Lee et al. [40] performed a cross-sectional analysis on Korean people with diabetes and found that HbA1c showed positive correlations with fasting and postprandial glucagon-to-insulin ratio. However, there were many differences between our study and Lee et al.’s study [40]. First, Lee et al. collected blood samples after an ad libitum diet instead of a standard meal, indicating that the dietary conditions were different between the study populations. Second, the glucagon-to-insulin ratio was used as the method of analysis, instead of a single indicator of glucagon; thus, it cannot be claimed that independent associations between glucagon and other variables were assessed. In addition, their study did not measure blood incretin levels. Therefore, it could be concluded that our study was more appropriate than Lee et al.’s study [40] for analyzing the correlations between postprandial changes in glucagon level and various factors. 

Our study had several limitations. First, blood glucagon and incretin levels were not measured continuously. The responses of various blood indicators to ingestion of food are generally assessed by the area under the curve of values repeatedly measured over a certain period or by a single value when the postprandial peak is reached. Theoretically, the area under the curve is more accurate than a peak value. However, we used only two measurements (basal and peak values) to minimize discomfort the subjects would experience due to repeated blood sampling. When people with diabetes ingest a mixed meal, blood glucagon and incretin are known to reach their peak level after approximately 30 min [7,10,11]. When we continuously measured glucagon, iGLP-1, and iGIP levels for 3 h after ingestion, the levels reached the peak after 30 min and gradually decreased thereafter (Appendix A). Accordingly, we set the blood sampling time for postprandial stimulation as 30 min. Second, we did not measure GLP-2. GLP-2 facilitates the proliferation of intestinal mucosal cells and promotes intestinal nutrient uptake, while it inhibits gastrointestinal motility and gastric acid secretion [41]. GLP-2 is co-secreted with GLP-1 from the L-cells in the intestinal epithelium, but, in contrast to GLP-1, GLP-2 stimulates glucagon secretion [42]. Some researchers believe that imbalance in glucagonotropic and glucagonostatic effects of gastrointestinal hormones, including GLP-2, GLP-1, and GIP, may be the cause for the inappropriate postprandial suppression of glucagon secretion [43]. Therefore, future studies are needed on the effects of GLP-2 on postprandial glucagon response. Third, we measured the intact form of incretin, but not the total form. The intact form is the biologically active form of incretin, whereas the total form includes inactive metabolites destroyed by DDP-4 [44]. Use of the total form is generally more advantageous for assessing the overall postprandial incretin secretion [45]. However, the intact form also accounts for secretion, but only to a certain extent [46] since most incretins secreted from the intestinal epithelium are destroyed before reaching glucagon-secreting pancreatic α-cells. Therefore, it can be viewed that the form of incretin associated with postprandial changes in glucagon levels is the intact form, not the total form. Fourth, most of the patients in our study population were taking antidiabetic medication. We excluded patients taking DDP-4 inhibitors or GLP-1 receptor agonists, which are known to affect blood glucagon levels. However, some studies have reported that metformin, insulin, and sulfonylurea could also affect glucagon levels, albeit rarely [47]. We performed multivariate analyses to eliminate the influence of diabetic medication taken by the patients, and the results confirm that metformin, insulin, and sulfonylurea are not factors that influence Δglucagon (Table 4). Fifth, the subjects of our study were not limited to patients with T2D. Inappropriate glucagon response to meal ingestion has been observed in almost all types of diabetes including T2D (type 1 diabetes [12], secondary diabetes to chronic pancreatitis [13] or pancreatectomy [14], and maturity-onset diabetes of the young [15]). In addition, diabetic patients in East Asia, including Korea, have a relatively severe decrease in insulin secretion [19,48], making it difficult to distinguish diabetes types. In our study, the significant relationship between Δglucagon and HbA1c was also observed when corrected with C-peptide, an insulin secretion index, in multivariate analysis.

In conclusion, glucagon secretion was not suppressed after the ingestion of a mixed meal in diabetes patients, and patients with poorer glycemic control showed a greater increase in postprandial glucagon levels. Moreover, such an inappropriate glucagon response was found to have no correlation with changes in the GLP-1 and GIP levels. The reason patients with poorer glycemic control show a greater increase in postprandial glucagon levels cannot be determined based on the findings in our study. Therefore, we believe that systematic experimental and clinical studies are needed in the future to investigate the interactions between glucagon and various gastrointestinal hormones including GLP-2.

## Figures and Tables

**Figure 1 jcm-10-02487-f001:**
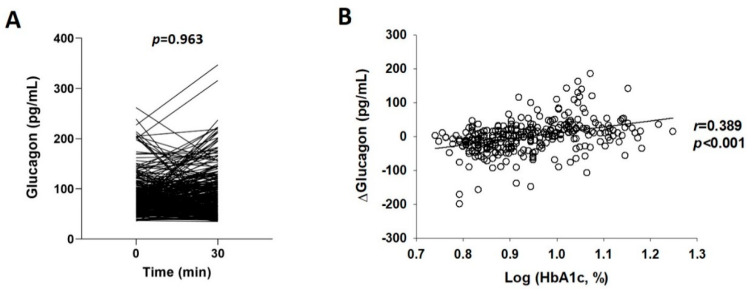
Changes in glucagon levels after ingestion of a standard mixed meal (**A**) and correlation between Δglucagon and HbA1c levels (**B**). HbA1c levels were logarithmically transformed. *p* values are calculated using the Wilcoxon signed-rank test (**A**) and *r* and *p* values are calculated using the Pearson’s correlation analysis (**B**). r, correlation coefficient; ΔGlucagon, (30-min postprandial glucagon level)—(basal glucagon level).

**Table 1 jcm-10-02487-t001:** Clinical and laboratory characteristics of the study subjects.

Variables	Value
*n*	317
Age, years	59.6 ± 11.5
Gender, male, *n* (%)	205 (64.6)
Body mass index, kg/cm^2^	25.8 ± 3.7
Waist circumference, cm	90.9 ± 9.0
Systolic BP, mmHg	138.1 ± 17.8
Diastolic BP, mmHg	81.9 ± 10.6
DM duration, years	9.9 ± 8.3
HbA1c, %	8.9 ± 2.2
Glucose, mg/dL	168.1 ± 66.5
C-peptide, ng/mL	2.1 ± 1.5
ALT, U/L	32.2 ± 26.4
Creatinine, mg/dL	1.1 ± 0.7
eGFR, mL/min	67.3 ± 17.3
Antidiabetic regimen	
Sulfonylurea, *n* (%)	169 (53.3)
Metformin, *n* (%)	221 (69.7)
Thiazolidinedione, *n* (%)	18 (5.6)
α-Glucosidase inhibitor, *n* (%)	18 (5.6)
Insulin, *n* (%)	100 (31.5)

Data are expressed as mean ± standard deviation or frequencies (%). BP, blood pressure; DM, diabetes mellitus; iGLP-1, intact glucagon-like peptide 1; iGIP, intact glucose-dependent insulinotropic polypeptide; ALT, alanine aminotransferase; eGFR, estimation of the glomerular filtration rate.

**Table 2 jcm-10-02487-t002:** Blood levels of glucose, C-peptide, glucagon, and intact incretin before and 30 min after ingestion of a standard mixed meal (*n* = 317).

Variables	Fasting Levels	30-min Post-Meal Levels	*p*
Glucose, mg/dL	168.1 ± 66.5	257.8 ± 71.2	<0.001
C-peptide, ng/mL	2.1 ± 1.5	3.2 ± 2.0	<0.001
Glucagon, pg/mL	84.6 ± 37.3	84.7 ± 48.2	0.909
iGLP-1, pmol/L	5.7 ± 3.7	11.5 ± 9.3	<0.001
iGIP, pmol/L	3.9 ± 3.8	21.8 ± 6.8	<0.001

Data are expressed as mean ± standard deviation. *p* values are calculated using Wilcoxon signed-rank test. iGLP-1, intact glucagon-like peptide 1; iGIP, intact glucose-dependent insulinotropic polypeptide.

**Table 3 jcm-10-02487-t003:** Correlational analysis of the relationships between Δglucagon levels and clinical and laboratory variables (*n* = 317).

Variables	ΔGlucagon, pg/mL	
	*r*	*p*
Age	0.005	0.925
Body mass index, kg/cm^2^	0.031	0.583
Waist circumference, cm	0.099	0.084
Systolic BP, mmHg	−0.042	0.456
Diastolic BP, mmHg	−0.039	0.485
Log (DM duration, years)	0.010	0.861
Log (HbA1c, %)	0.389	<0.001
Log (Fasting glucose, mg/dL)	0.198	<0.001
Log (Fasting C-peptide, ng/mL)	−0.052	0.360
ΔGlucose, mg/dL	−0.269	<0.001
ΔC-peptide, ng/mL	−0.036	0.531
Log (ΔiGLP-1, pmol/L)	0.09	0.118
Log (ΔiGIP, pmol/L)	−0.009	0.869
Log (ALT, U/L)	0.025	0.667
Log (Creatinine, mg/dL)	0.053	0.351
Log (eGFR, mL/min)	−0.140	0.013

DM duration, HbA1c, fasting glucose, C-peptide, ΔiGLP-1, ΔiGIP, ALT, creatinine, and eGFR were logarithmically transformed. *r* and *p* values are calculated using the Pearson correlation analysis. *r*, correlation coefficient; BP, blood pressure; DM, diabetes mellitus; iGLP-1, intact glucagon-like peptide 1; iGIP, intact glucose-dependent insulinotropic polypeptide; ALT, alanine aminotransferase; eGFR, estimation of the glomerular filtration rate; Δ, difference between the 30-min postprandial value and the basal value.

**Table 4 jcm-10-02487-t004:** Hierarchical multiple regression analyses predicting Δglucagon levels.

Variables	Model 1		Model 2		Model 3	
	*β*	*p*	*β*	*p*	*β*	*p*
Age, years	−0.028	0.622	−0.056	0.319	−0.049	0.391
Gender, female	0.125	0.019	0.105	0.050	0.109	0.041
Log (HbA1c, %)	0.359	0.000	0.337	0.000	0.327	0.000
Log (eGFR, mL/min)	−0.099	0.082	−0.037	0.527	−0.014	0.821
Log (Fasting glucose, mg/dL)	0.007	0.905	0.040	0.563	0.027	0.703
Log (Fasting C-peptide, ng/mL)	0.002	0.971	−0.030	0.628	−0.005	0.943
ΔGlucose, mg/dL			-0.237	0.000	−0.222	0.000
ΔC-peptide, ng/mL			0.092	0.131	0.087	0.156
Log (ΔiGLP-1, pmol/L)			0.032	0.573	0.027	0.632
Log (ΔiGIP, pmol/L)			0.067	0.228	0.062	0.262
Use of sulfonylurea					−0.044	0.463
Use of metformin					−0.058	0.312
Use of insulin					0.023	0.716
Adjusted R^2^	0.175		0.243		0.250	
F	10.874	<0.001	9.037	<0.001	7.127	<0.001

HbA1c, eGFR, fasting glucose, C-peptide, ΔiGLP-1, and ΔiGIP were logarithmically transformed. *β*, corrected regression coefficient; eGFR, estimation of the glomerular filtration rate; iGLP-1, intact glucagon-like peptide 1; iGIP, intact glucose-dependent insulinotropic polypeptide; Δ, difference between the 30-min postprandial value and the basal value.

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
