# Peer review of "The Change in Glucagon Following Meal Ingestion Is Associated with Glycemic Control, but Not with Incretin, in People with Diabetes"

_jcm, 2021, doi:10.3390/jcm10112487_

Round 1
Reviewer 1 Report
In the paper entitled “The Change in Glucagon Fllowing Meal Ingestion is Associated with Glycemic Control,but not with Incretin, in People with Diabetes" Soyeon Yoo and coworkers investigated changes in glucagon levels in postprandial condition in diabetic patients.
By analyzing a large cohort of diabetic patients, Authors conclude that postprandial glucagone levels are not correlate with incretin levels changing and consequently those class of metabolic hormones seems not to be implied in postprandial glucagon release.
The paper is well written, rationale is clear and figure and tables are exhaustive.
Nevertheless, some important concerns are to be assessed to the Authors:
- The background in Abstract section (row 11) needs to be rewritten since Authors reported the aim of the study instead of background.
- Paragraph titles (row 154, 182, 204) should report at least a data summary of authors’ finding; it is useless to report just a simple description of analysis performed in the section. Authors better modify the paragraph titles with a brief description of data they obtained.
- In figure 1 Authors clearly demonstrated that glucagon postprandial levels do not change after a standard mixed meal ingestion in their cohort, however in row 186 Authors reported that glucagon variation show a positive correlation with HbA1c and no correlation among iGLP-1 and iGIP levels (row 215). According to figure 1, which glucagon variation are the Authors refer to? This is a crucial point that need to be better explained, as this is the main finding of entire paper.
- In discussion section, row 280 could the Authors explain why they expected a negative correlation among iGLP-1 and glucagon variation.
Furthermore, Authors can consider some minors concerns
- In title, row 2 authors should correct the type in the word Fllowing
- Referring to changing of glucagon level reported all over the manuscript text, Authors should change the greek symbol D whit the word “variation”
Author Response
We thank the editor and reviewers of Journal of Clinical Medicine for taking their time to review our article. We have made some corrections and clarifications in the manuscript after going over the reviewer’s comments. The changes are summarized below:
(The changes in the manuscript are marked in red.)
<Reveiwer1>
In the paper entitled “The Change in Glucagon Fllowing Meal Ingestion is Associated with Glycemic Control,but not with Incretin, in People with Diabetes" Soyeon Yoo and coworkers investigated changes in glucagon levels in postprandial condition in diabetic patients.
By analyzing a large cohort of diabetic patients, Authors conclude that postprandial glucagone levels are not correlate with incretin levels changing and consequently those class of metabolic hormones seems not to be implied in postprandial glucagon release.
The paper is well written, rationale is clear and figure and tables are exhaustive.
Nevertheless, some important concerns are to be assessed to the Authors:
- The background in Abstract section (row 11) needs to be rewritten since Authors reported the aim of the study instead of background.
Thank you for your valuable comments.
As you pointed out, in our paper, the purpose of the study is mainly described in the background part of Abstract. It would be nice to describe additional contents in the background, but please understand that it is not possible due to the limit on the number of characters in the abstract.
In addition, when looking at the explanation of JCM's Instructions for Authors- Manuscript Preparation-Front Matter-Abstract- background section, the description of the purpose of the study is also allowed.
“Instructions for Authors- Manuscript Preparation-Front Matter-Abstract- Background: Place the question addressed in a broad context and highlight the purpose of the study.”
- Paragraph titles (row 154, 182, 204) should report at least a data summary of authors’ finding; it is useless to report just a simple description of analysis performed in the section. Authors better modify the paragraph titles with a brief description of data they obtained.
Thanks for the good comment. As you pointed out, the paragraph titles have been changed.
3.1. Clinico-biochemical characteristics of subjects and changes in incretin and glucagon levels before and after ingestion of the mixed meal
--> 3.1. Clinico-biochemical characteristics of subjects and ingestion of mixed meal induce increase incretin level, but not glucagon levels.
3.2. Univariate analyses of the relationship of Δglucagon levels with clinico-biochemical variables
--> 3.2. In univariate analysis, Δglucagon levels showed correlation with HbA1c, fasting glucose, Δglucose and GFR, but not incretin levels.
3.3. Hierarchical multiple regression analysis predicting Δglucagon levels.
--> 3.3. In hierarchical multiple regression analysis, HbA1c was the variable predicting Δglucagon levels.
- In figure 1 Authors clearly demonstrated that glucagon postprandial levels do not change after a standard mixed meal ingestion in their cohort, however in row 186 Authors reported that glucagon variation show a positive correlation with HbA1c and no correlation among iGLP-1 and iGIP levels (row 215). According to figure 1, which glucagon variation are the Authors refer to? This is a crucial point that need to be better explained, as this is the main finding of entire paper.
Thank you for your valuable comments.
As shown in Table 2, there was no significant difference in the mean value of glucagon levels before and after mixed meal ingestation in the subjects of this study. As described in the introduction, after ingesting a meal, healthy adults show suppressed glucagon secretion, whereas diabetes people do not show suppression of glucagon secretion. In Figure 1A, glucagon changes according to mixed meal ingestation of each individual vary. In addition, glucagon secretion is not only affected by meal ingestation, but also by various factors such as insulin secretion, blood sugar, and incretins.
In this study, we tried to analyze the factors affecting inapproprate glucagon secretion in diabetic patients. Therefore, the relationship between various factors and the amount of change in glucagon level (Δglucagon = 30min postprandial glucacon – basal glucagon) according to mixed meal ingestation in each subject was analyzed.
When various factors that may affect glucagon secretion were corrected and analyzed (Table 3, Table 4), Δglucagon showed positive correlation with HbA1C, but not with incretin.
Therefore, this study concluded that glucagon suppression does not occur after meal ingestion in diabetic patients when simply comparing mean value of glucagon levels. However, in each individual, as blood glucose control is poor, the greater the amount of change in glucagon level (Δglucagon) caused by meal ingestion, which is not related to incretin.
- In discussion section, row 280 could the Authors explain why they expected a negative correlation among iGLP-1 and glucagon variation.
Thank you for your valuable comments. As described in row 277, when intravenous infusion of GLP-1, glucagon secretion was decreased. In general, GLP-1 is known to inhibit glucagon secretion by acting on islet alpha cells of pancreas. Therefore, a negative correlation was expected, but this correlation was not shown in our data. As described later, similar results have been reported by other researchers, and it is believed that changes in glucagon levels due to meal ingestation in diabetic patients are not caused by endogenous GLP-1 or GIP.
Furthermore, Authors can consider some minors concerns
- In title, row 2 authors should correct the type in the word Fllowing
Thank you for accurately pointing out our mistake. Has been modified.
Fllowing --> Following
- Referring to changing of glucagon level reported all over the manuscript text, Authors should change the greek symbol D whit the word “variation”
In “2. Materials and methods - 2.2. Study procedure”, definition of the greek symbol D (Δ; delta) was explained (Row 111~113). “Δ” means for the difference between the 30-min postprandial value and the basal value. As it may cause confusion, explanations for Δ have been added to the bottom of Figure 1, Table 3, and Table 4 (row 181~182, row 205~206, row229)

Reviewer 2 Report
The manuscript entitled “The Change in Glucagon Following Meal Ingestion is Associated with Glycemic Control, but not with Incretin, in People with Diabetes” aims at understanding glucagon profile and effects after 30 minutes of ingesting a meal tolerance test.
Major comments
- The authors report a meal tolerance test (MTT) just for the first 30 minutes. Is there a reason to not report the full MTT? One cannot exclude that glucagon suppression is not delayed in subjects with diabetes.
- The authors do not report any insulin data, would be quite interesting to refer to the insulin values specifically for the individuals that are not having insulin therapy. I would advise dividing the groups especially in 2; the ones that are not exposed to insulin therapy and the ones that are exposed. Also, report within the groups the subjects with high and low glycemic control.
- Regarding the group that is not exposed to exogenous insulin a better assessment for the glucagon efficacy would be the ratio of secreted insulin (C-peptide) to glucagon ratio. The ratio will help to interpret the capacity to shut down or not the hepatic glucose production.
- Moreover, the group that is not exposed to exogenous insulin can be further characterized for insulin resistance, hepatic insulin resistance, adipose tissue insulin resistance and b cell function. The interesting finding of the paper is that in individuals with diabetes glucagon levels are not suppressed with a meal in the first 30 minutes which is associated with HbA1c but not with GLP-1 or GIP. The resultant increase in glycemia can be a result of an increase in hepatic glucose production and/or increased insulin resistance promoted by glucagon. Please include the latter mechanism in the discussion as it is supported by Patarrão et al (Patarrão, R. S., & Macedo, M. P. (2015). Acute glucagon induces postprandial peripheral insulin resistance. PLoS ONE, 10(5)).
- The authors refer to table 3 and 4 but I could not find them in the manuscript. Thereafter it is difficult to interpret the results.
- The authors should be careful with their exclusive interpretation that “elevated HbA1c indicates reduced ability to secrete insulin”. Indeed, other mechanisms can explain the increased glucose levels for example the increased glucagon levels and therefore an increase in hepatic glucose output and increased insulin resistance.
Minor comments
- In the title the word following is incorrect.
- It is recommended that the authors describe the meal tolerance test in the methods section.
Round 2
Reviewer 2 Report
Dear authors please reply adequately for the following comments:
Regarding former comment 1)
As you reply:
“
As described in the Introduction, after ingesting a mixed meal, blood glucagon levels become significantly higher in diabetes people than in healthy individuals, and the peak level is reached within 30 minutes after ingestion. In about 40 diabetic patients, when we continuously measured glucagon, iGLP-1, and iGIP levels for 3 hours after mixed meal ingestion; the levels reached the peak after 30 minutes and gradually decreased thereafter (data not shown).”
Please add the data not shown as a figure. Could be considered as a supplementary figure or panel a)
Regarding the former comment 3, 4 and 6 the authors reply was:
“This study focused primarily on the change of glucagon secretion in islet cells caused by mixed meals in diabetic patients rather than glucagon action. Therefore, the analysis was focused on the amount of change of glucagon rather than the insulin to glucagon ratio”
“ This study was conducted in Koreans, and diabetic patients in Far East Asia, including Korea, have a relatively severe decrease in insulin secretion (Hence, in Koreans, the main cause of the increase in blood sugar is lower insulin secretion rather than insulin resistance in long standing DM.) In this study subjects, the average duration of diabetes was about 10 years, and the BMI was 25.8%.”
and
“This study was conducted in Koreans, and diabetic patients in Far East Asia, including Korea, have a relatively severe decrease in insulin secretion (Hence, in Koreans, the main cause of the increase in blood sugar is lower insulin secretion rather than insulin resistance in long standing DM.) In this study subjects, the average duration of diabetes was about 10 years, and the BMI was 25.8%.”
I am fully aware that Korean have a decreased b cell capacity. Nevertheless, one should not ignore other mechanisms. Moreover, the authors have the insulin levels. Could you please show the insulin levels and make the other calculations. Adjust your discussion accordingly.
